# The Biochemical Diagnosis of Acromegaly

**DOI:** 10.3390/jcm10051147

**Published:** 2021-03-09

**Authors:** Amit Akirov, Hiba Masri-Iraqi, Idit Dotan, Ilan Shimon

**Affiliations:** 1Institute of Endocrinology, Beilinson Hospital, 49100 Petach Tikva, Israel; HibaMa@clalit.org.il (H.M.-I.); IDITDO@clalit.org.il (I.D.); Ilanshi@clalit.org.il (I.S.); 2Sackler School of Medicine, Tel Aviv University, 39040 Tel Aviv, Israel

**Keywords:** acromegaly, growth hormone, IGF-1, pituitary

## Abstract

Background: The diagnosis of acromegaly still poses a clinical challenge, and prolonged diagnostic delay is common. The most important assays for the biochemical diagnosis and management of acromegaly are growth hormone (GH) and insulin-like growth factor-1 (IGF-1). Objective: Discuss the role of IGF-1, basal serum GH, and nadir GH after oral glucose tolerance test (OGTT) for the diagnosis, management, and treatment of patients with acromegaly. Methods: We performed a narrative review of the published data on the biochemical diagnosis and monitoring of acromegaly. An English-language search for relevant studies was conducted on PubMed from inception to 1 January 2021. The reference lists of relevant studies were also reviewed. Results: Serum IGF-1 levels, basal GH values, and nadir GH after OGTT play a major role in the diagnosis, management, and treatment of patients with acromegaly. Measurement of IGF-1 levels is the key factor in the diagnosis and monitoring of acromegaly, but basal and nadir GH following OGTT are also important. However, several factors may significantly influence the concentrations of these hormones, including assay methods, physiologic and pathologic factors. In some cases, discordant GH and IGF-1 levels may be challenging and usually requires additional data and monitoring. Conclusion: New GH and IGF-1 standards are much more precise and provide more accurate tools to diagnose and monitor patients with acromegaly. However, all these biochemical tools have their limitations, and these should be taken under consideration, along with the history, clinical features and imaging studies, when assessing patients for acromegaly.

## 1. Introduction

Acromegaly is a slowly progressive disease caused by persistent excess of circulating growth hormone (GH) and insulin-like growth factor-1 (IGF-1) [1]. While most cases are secondary to a GH-secreting pituitary adenoma, acromegaly may rarely be secondary to a hypothalamic secreting GH-releasing hormone (GHRH) or ectopic GHRH or GH secretion. High levels of IGF-1 are responsible for most of the clinical manifestations of acromegaly [2,3].

More than 130 years after the French neurologist Pierre Marie coined the term “l acromegalie”, the diagnosis of acromegaly, which is based on clinical and biochemical findings, still poses a clinical challenge.

Some patients are now presenting with no clinical features at all of acromegaly, in others, the clinical features may be subtle in the early stages of the disease and the slowly progressing changes in physical features may go unnoticed by the patients and their family and friends, hence, the diagnosis is frequently preceded by approximately 4–10 years of unrecognized active disease and prolonged diagnostic delay is associated with increased morbidity and mortality [4,5].

Female patients have more prolonged diagnostic delays compared to males [6]. While over the years there have been improvements in the diagnostic procedures for acromegaly, including the availability of serum IGF-1 measurement and more sensitive GH assays, the diagnostic delay and the age at diagnosis have not changed, and that is mostly secondary to the insidious onset of the clinical features, overlap of the signs and symptoms with other common conditions, and lack of disease awareness among other medical specialists [5,7]. However, this diagnostic delay is considerably shorter compared with mean delays in diagnosis of 16–18 years after symptom onset reported half a century ago [8].

The most important assays for the diagnosis and management of acromegaly are GH and IGF-1 levels. The 2014 Endocrine Society Clinical Practice Guideline on acromegaly recommend measurement of IGF-1 levels in patients with typical clinical manifestations of acromegaly, especially the classical acral and facial features, or in patients who have several of the associated conditions, including sleep apnea, hypertension, type 2 diabetes mellitus, arthritis, carpal tunnel syndrome, and hyperhidrosis. The recommended strategy is to measure serum IGF-1 and in patients with elevated or equivocal serum IGF-1 levels, confirming the diagnosis by demonstrating an unsuppressed GH value ≥ 1 μg/L following documented hyperglycemia during an oral glucose tolerance test (OGTT) [9]. Subsequently, this cut-off was revised to 0.4 μg/L when using ultrasensitive GH assays [10,11].

Measurement of GH and IGF-1 levels is also used to determine surgical remission, response to medical treatment, and guide treatment decisions. As IGF-1 levels measured at 12 weeks after surgery reflect clinical activity of disease, biochemical remission following surgery is defined as normal IGF-1 levels and GH < 0.4 μg/L [9]. However, according to a recent consensus statement on multidisciplinary management of acromegaly IGF-1 values up to 1.2–1.3× upper limit of normal (ULN) range may be considered sufficient for control of acromegaly [12].

While serum IGF-1 levels, basal GH values, and nadir GH after OGTT play a major role in the diagnosis, management, and treatment of patients with acromegaly, the limitations of these tests should be acknowledged for optimal use.

## 2. Insulin-Like Growth Factor-1

In contrast to the variability of GH, IGF-1 is secreted continuously, has a longer half-life and exhibits more stable concentrations in blood, and levels are proportionate to the mean GH levels, although this correlation between GH with IGF-1 may be lost at higher GH values, possibly due to saturation of hepatic GH receptors [8,13]. Furthermore, IGF-I levels can be determined at any time of the day without the patient having to fast [14]. For these reasons, measuring IGF-1 level is the initial test recommended in patients with clinically suspected acromegaly and IGF-1 levels below normal upper range for age and sex can effectively exclude the diagnosis of acromegaly [1,9]. In patients with clinical features suggestive of acromegaly, elevated IGF-1 is usually sufficient to complete the diagnosis, and the OGTT may be used as a confirmatory test [15].

While IGF-1 is synthesized primarily in the liver, additional tissues can produce IGF-1, including bone, muscle, kidney and the pituitary gland itself. In the serum, less than 1% of total serum IGF-1 circulates as a free hormone, whereas most IGF-1 is bound to IGF-1 binding protein (IGFBP-3) or IGFBP-5, and acid-labile subunit [1].

Several factors may influence IGF-1 assays and result in wide variability between assays, including the number of normal subjects that are adequate to establish age-adjusted reference ranges, the affinity and specificity of the antibodies that are used by each assay, and the method of removing binding protein interferences [16]. Porkajac and colleagues studied GH and IGF-1 assay performance close to cut-off values for active acromegaly and submitted serum samples known to give borderline results to 23 centers for IGF-1 assessment and reported more than a 2-fold variation in IGF-1 cut-off for the ULN range measured in different laboratories and the diagnosis of acromegaly was inaccurately excluded in 30% of samples assayed for IGF-1 [17]. Assays to determine IGF-1 and GH levels include electrochemiluminescence immunoassay (Roche Cobas) or chemiluminescence immunoassay (Siemens Immulite, IDS-iSYS, Diasorin Liaison), which introduce antibodies to bind and separate out target antigens, or liquid chromatography-mass spectrometry, which selects the target analyte based on mass and use an electric ion field to determine the values, and there could be notable discrepancies in IGF-1 test results between different assays [13,14]. Therefore, the recommendation is to use the same well-validated IGF-1 assay throughout patient follow-up [12].

Several physiologic factors may alter IGF-1 concentrations. Severe obesity, malnutrition, and prolonged fasting can reduce IGF-1 levels in patients with and without acromegaly [18,19]. High GH with low IGF-1 can be observed in states of GH resistance such as systemic inflammation, chronic liver disease, cirrhosis, and anorexia nervosa [20,21]. Oral estrogen and selective estrogen receptor modulators (SERM) can also render the liver less responsive to GH, resulting in lower IGF-1 levels. However, transdermal estrogens have no impact on IGF-I levels [22]. Renal failure can be associated with higher GH levels but IGF-1 values may remain unchanged or even decrease [23] (Table 1).

Age is an important physiologic factor that may alter IGF-1 levels, as values rise from birth until puberty, followed by a decline with advancing age [19]. During pregnancy, the pituitary GH is suppressed due to the significant increase in placental GH secretion that induces hepatic IGF-1 secretion and IGF-1 levels increase 2- to 3-fold in the second half of pregnancy, with a peak at around 37 weeks [24]. Jallad and colleagues provided data on the outcomes of pregnancies of women with acromegaly, reporting that IGF-1 normalized during pregnancy in 13 of 15 patients followed without any specific treatment. In most women, IGF-1 elevation was reported following delivery [25].

Treatment is aimed at normalizing IGF1 levels, which reflects adequate disease control and decreases risk of developing complications. The 2014 Endocrine Society Clinical Practice Guideline on acromegaly suggest measuring IGF-1 levels at 12 weeks or later after pituitary surgery for acromegaly and, as IGF-1 level reflect clinical activity of acromegaly, this measure is used to guide the need for additional treatments, including reoperation, medical treatment, or radiation therapy, and control of GH and IGF-1 excess can reduce morbidity and mortality [9]. According to a recent consensus statement on multidisciplinary management of acromegaly IGF-1 values up to 1.2–1.3× ULN range may also be considered sufficient for control of acromegaly [12].

## 3. Basal Growth Hormone

Basal GH levels are measured in the morning after fasting and are usually elevated in patients with acromegaly [8].

Basal serum GH levels and nadir GH following OGTT are higher in younger, pre-menopausal women, compared with older women. In addition, basal serum GH levels in younger women are highest during mid-cycle and change with phases of the menstrual cycle [26].

Somatotroph cells in the anterior pituitary gland release GH under hypothalamic control through GHRH, which increases transcription of the GH gene and synthesis and secretion of GH [8]. The release of GH is pulsatile, with 6–11 pulses of GH secretion within a 24-h period, while for 70–80% of the day, GH levels remain below the level of detection [8]. In patients with acromegaly, basal serum GH levels are continuously elevated, but wide fluctuations are common due to inherent episodic GH secretion from both the normal and GH-secreting pituitary adenoma [27].

Elevated basal GH levels are not specific for acromegaly, and levels may be significantly elevated in poorly controlled diabetes mellitus secondary to reduction in somatostatin release, renal failure associated with increased GH release and reduced GH clearance, malnutrition, in the setting of stress or during exercise and sleep [20,26]. In pregnancy, measurement of GH is challenging due to homology between GH and placental GH [28] (Table 1).

Measuring fasting GH on the first couple of days following pituitary surgery for acromegaly may provide an early indication of remission with sensitivity rates of 65–97% and specificity rate of 77–93% [29]. Serum GH values can be used to assess and monitor disease control following surgery and with adjuvant medical therapy with somatostatin receptor ligands, with the goal of achieving a fasting level <1.0 μg/L [12].

## 4. Nadir Growth Hormone after OGTT

In normal subjects, glucose loading results in suppressed GH level, most likely due to inhibition of GHRH and/or stimulation of somatostatin release. According to the 2014 Endocrine Society Clinical Practice Guideline on acromegaly, lack of suppression of GH following 75 g OGTT is the gold standard diagnostic test for acromegaly and this should completed as a confirmatory test in patients with elevated or equivocal serum IGF-1 levels [9]. The guidelines suggest using GH nadir < 1µg/L after OGTT to exclude the diagnosis of acromegaly, although when using highly sensitive GH assay, a lower GH nadir of 0.4 µg/L has been proposed [9,15].

While it was previously considered as the gold standard for the diagnosis of acromegaly, suppression of GH levels during OGTT is not specific for acromegaly, as conditions associated with an abnormal GH response to a glucose load include severe catabolic illness, malnutrition, renal or hepatic failure, uncontrolled diabetes mellitus, Laron type dwarfism. In addition, abnormal GH response was also reported in patients treated with L-dopa and those who use heroin. However, as IGF-1 levels in most of these conditions are not elevated, the diagnosis of acromegaly can be excluded [8,16] (Table 1).

In patients with abnormal glucose metabolism and diabetes, nadir GH following OGTT was previously reported to be an unreliable diagnostic test with a lower specificity and increased false positive results, since GH levels may not suppress normally after glucose load [16]. Insulin induces expression of GH receptors on hepatocytes and may have an effect at the post-receptor level; hence, portal hypoinsulinemia is associated with GH resistance resulting in compensatory elevated GH [17]. In certain patients, a paradoxical increase of GH levels after glucose load was reported, most likely mediated by glucose-dependent insulinotropic polypeptide (GIP), a peptide released after oral glucose load [18]. Paradoxical GH response after OGTT may reflect the pathophysiological severity of acromegaly and may predict the effectiveness of treatment with somatostatin receptor ligand [19]. The paradoxical response of GH to OGTT was also reported in patients with impaired glucose tolerance or diabetes, anorexia nervosa, puberty, renal or liver failure, or malnutrition [20] (Table 1).

While GH measurements in earlier studies were based on conventional radioimmunoassay with limited sensitivity, more recent studies used modern GH assays to evaluate GH suppression and showed no difference in nadir GH levels following OGTT [21,22]. Dobri and colleagues recently reported that with modern highly sensitive GH assays, a nadir GH ≥ 0.4 µg/L following OGTT may be used for the diagnosis of acromegaly in patients with impaired fasting glucose, impaired glucose tolerance, or relatively well-controlled diabetes mellitus with glycated hemoglobin <8% [22]. Additional studies are required to determine the role of nadir GH following glucose load in patients with poorly controlled diabetes, and there is no consensus on the role of nadir GH after OGTT for diagnosis of acromegaly in these patients. At this point, experts agree to repeat measurement after improvement of glycemic control [23].

Normalization of IGF-1 is considered the primary criterion for establishing remission of acromegaly and the role of nadir glucose following glucose load in the monitoring of patients following surgery is less clear [9,12]. Data suggest that while normalization of IGF-1 can be delayed up to 12 months after surgical intervention for acromegaly, normalization of GH nadir can be seen as early as 1 week after surgery [24]. The 2014 Endocrine Society Clinical Practice Guideline on acromegaly suggests measuring a nadir GH level after a glucose load in a patient with a random serum GH > 1 g/L [9]. According to an expert consensus document on acromegaly therapeutic outcomes, GH nadir levels <1 µg/L after glucose load reflect cure following surgical intervention, with evidence to support GH suppression is associated with improved long-term outcomes and reduced mortality. Again, with ultra-sensitive GH assays, the recommended GH cut-off is 0.4 µg/L [11,25]. Importantly, GH levels may be impacted by preexisting use of somatostatin receptor ligands which are recommended as first-line medical treatment and, therefore, measuring GH nadir following OGTT is not likely to be clinically useful in patients treated with somatostatin receptor ligands [26,27].

## 5. Discordant GH and IGF-1 Values

Management of acromegaly is challenging in patients with discordant findings obtained from measurement of GH and IGF-1 levels.

### 5.1. Elevated IGF-1, Normal GH

Micromegaly is a term coined by Diamarki et al. in 2002, and has been used to describe patients with clinical features of acromegaly associated with elevated IGF-1 levels, but normal basal GH levels, and in many cases normal nadir GH following glucose load [28]. Ribeiro and colleagues reported that of 40 newly diagnosed patients with acromegaly, 33% had a GH nadir <1 μg/L, and 18% had a GH nadir <0.4 μg/L [29]. Espinosa and colleagues recently reported that normal basal GH is uncommon in patients with acromegaly, and these patients were significantly older, more likely to be male, harbor small and non-invasive microadenomas, with lower IGF-1 levels and nadir GH following OGTT, and yet had a similar disease duration and a similar clinical course to patients with elevated basal GH at diagnosis [30]. Hence, in patients suspicious for acromegaly, discordant GH and IGF-1 values with elevated IGF-1 levels but normal GH suppression following OGTT, may not definitively exclude the diagnosis. On the other hand, in those with elevated IGF-1 levels but low clinical suspicion for acromegaly, continued follow-up is recommended [31]. Rosario and colleagues reported the evolution of 42 patients with a clinical scenario suggestive of acromegaly, elevated IGF-1 levels and normal nadir GH following OGTT. Five years after the initial evaluation, 15 patients had normal IGF-1 levels, while 27 patients had persistently elevated IGF-1 levels and all had normally suppressed GH following additional OGTT. Imaging revealed a potential pituitary microadenoma in 2 patients, and acromegaly was ruled out in 40 patients. These findings suggest that acromegaly is very unlikely in patients with suppressed GH after glycemic load [32].

Butz and colleagues emphasized that while IGF-1 levels are the key factor in the diagnosis and monitoring of acromegaly, GH measurements should not be ignored, as these two hormones reflect different aspects of the disease: GH is a measure of the secretory activity of the tumor and provides a correlate of pituitary GH secretion, while IGF-1 is a measure of biochemical and biological activity of the disease and the peripheral response to circulating GH [33,34].

### 5.2. Normal IGF-1, Elevated GH

In the absence of clinical conditions that reduce serum IGF-1 levels—including diabetes mellitus, thyroid, kidney or liver disease, anorexia nervosa, weight loss >5% in the last three months and body mass index <18.5 kg/m^2^, and oral estrogens—normal IGF-1 levels should rule out acromegaly, even with basal and nadir GH levels following GH suppression test >0.4 μg/L. Rosasrio and colleagues identified 179 patients with extremity enlargement, and following exclusion of patients with elevated IGF-1 and those with conditions that would interfere with laboratory investigation, determined basal GH levels, and in those with basal GH > 0.4 μg/L, completed GH suppression test. Basal GH > 0.4 μg/L was observed in 103 patients and nadir GH > 0.4 μg/L was found in 21 (30%) subjects, that were submitted to MRI, which revealed microadenoma in only two patients. These two patients continued to have normal IGF-1 values in subsequent measurements and no clinical progression was reported after 4 years [35].

Several key points should be considered in cases of discordant GH and IGF-1 values:If there is a significant discrepancy between GH and IGF-1 levels, repeat testing may be warranted, particularly if the clinical scenario is suggestive of acromegaly.Diagnosis of acromegaly in pregnancy is confounded by GH secretion by the placenta and increased IGF-1 values, as available commercial assays are not able to discriminate between the source of GH secretion [36].Acute critical illness can cause physiological higher peak in GH secretion and low concentrations of IGF-1 and when acromegaly is suspected, testing should be done after recovery [37].In patients on oral estrogens, GH levels may be falsely elevated, and it is recommended to stop the estrogen for three months before retesting [38].Low IGF-1 in the presence of clinical acromegaly could represent a later stage of a disease process that was initially associated with elevated IGF-1, resulting in burnt-out acromegaly, which may be associated with other features of hypopiuitarism [14].Uncontrolled diabetes may be associated with low IGF-1 levels and hamper the diagnosis of acromegaly, but a rise in IGF-1 levels following glycemic control may support the diagnosis [23].

## 6. Additional Tests

There are additional tools that may be used to aid in the assessment of patients with suspected acromegaly. However, these tests are not commonly used and are not recommended by professional societies for the confirmation of the diagnosis of acromegaly.

24-h serum GH profile—frequent serum GH sampling may overcome the limitation associated with the pulsatile nature of GH secretion resulting in wide fluctuations in random GH measurement. However, this method for assessing GH secretion is not practical in most settings [39,40].Urinary GH—assessment of 24-h mean urinary GH levels correlates with random serum GH levels, IGF-1 levels, and clinical activity of acromegaly, but as it rarely provides any additional information, it is usually reserved for research purposes [8,39].IGFBP-3—the most abundant IGF-1 binding protein that binds 80–90% of circulating IGF-1 in a trimeric complex that also includes the acid-labile subunit. IGFBP-3 levels may assist in the diagnosis and monitoring of acromegaly as there is a direct correlation between IGFBP-3 levels with IGF-1 [41].GHRH—measurement of GHRH will be reserved for patients with suspected ectopic GHRH secretion causing acromegaly, for instance, in patients with no evidence of a pituitary adenoma on imaging [8,23].

## 7. Conclusions

With the emergence of the ultrasensitive assays, the present GH and IGF-1 assays in combination with the new GH and IGF-1 standards are much more precise and provide more accurate tools to diagnose and monitor patients with acromegaly. However, these biochemical tools have their limitations, and these should be taken under consideration, along with the history, clinical features, and imaging studies, when assessing patients for acromegaly.

## Figures and Tables

**Table 1 jcm-10-01147-t001:** Conditions affecting serum insulin-like growth factor-1 (IGF-1) and growth hormone (GH) levels. Several physiologic and pathologic conditions may significantly influence IGF, basal, and nadir GH levels following glycemic load.

Condition	IGF-1	Basal GH	Nadir GH
Puberty	High	High	High
Pregnancy	High *	High	High
Diabetes Mellitus	Low/Normal	High	High
Renal Failure	Low/Normal	High	High
Liver Disease	Low/Normal	High	High
Malnutrition/Anorexia	Low/Normal	High	High
Oral Estrogen	Low/Normal	High	High
Critical Illness	Low/Normal	High	High

* In pregnant women with known acromegaly, IGF-1 levels may normalize spontaneously during pregnancy.

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
