# Peer review of "The Biochemical Diagnosis of Acromegaly"

_jcm, 2021, doi:10.3390/jcm10051147_

Round 1

Reviewer 1 Report

This review article by Akirov et al. describes how to diagnose acromegaly.

The article is well written and will be a good read for generalists and even endocrinologists who do not see many acromegaly patients. The review article is laid out and flows well.

Minor points:

  1. Suggest that the authors devote and elaborate more of the issue of of Discordant GH and IGF-I values. Obviously there is a higher likelihood that clinicians will come across more patients with raised IGF-I and normal GH vs patients with normal IGF-I and raised GH because if one is suspecting acromegaly, IGF-I is usually performed first. Perhpas the authors could provide a table with these two scenarios and provide bullet points to clinicians how to manage these patients with these two discordant scenarios.
  2. The authors are encouraged to provide a paragraph to discuss the pitfalls and limitations of GH and IGF-I assays as this can be a cause of discordant GH and IGF-I values. Also, with assay heterogeneity in mind, the authors can provide recommendations to clinicians how to overcome these limitations so as not to cause potential confusion when faced with discordant results.
  3. In Section 6 of the paper, the use of 24 hr serum GH profile, urinary GH, IGFBP3 and GHRH are not commonly used tests and this needs to be emphasized. Morepver these tests are not rececommended by professional societies when confirming the diagnosis.   

Author Response

  1. Thank you for this comment. we have provided bullet points for clinicians, in addition to Table 1 (page 6, lines 261-280; Table 1).
  2. We have elaborated on the available GH and IGF-1 assays, and recommend to use the same well-validated IGF-1 assay throughout patient follow-up (page 2, last line until page 3, line 103).

  3. We agree with this comment and now emphasize this (page 6, lines 284-285).

Reviewer 2 Report

Extremely well written and clear manuscript and summary. My only comment is to also make a small mention in the introduction and in 5.1 on discordant results about 1) that some patients are now presenting with no clinical features at all of acromegaly and 2) That suppressed GH on OGTT definitely does not exclude acromegaly (this is eluded to about "micromegaly', but can even occur in macroadenomas. In my personal experience, there appears to be a surge of cases of acromegaly cases with no clinical features. I have recently had two MACROADENOMAS with no clinical features. Both had mildly raised IGF-1, but one fully suppressed on OGTT, the other didn't suppress. The histology stained for Pit-1 confirming both were somatoropinomas.

Author Response

Thank you for your comments, we have added the suggested comments to the manuscript (page 1, lines 37-38 & page 5, lines 228-230) .

Reviewer 3 Report

The authors reviewed the role of GH (OGTT), IGF-1 measurement in the management of acromegaly mainly focusing on their limitations. Also, they covered discrepancies between IGF-1 and OGTT. They accommodated subtle changes after consensus meeting in 2019 (Giustina et al. in 2020) in an adequate manner. 

As this is a review article, they may not be able to provide suggestions how to harness IGF-1 measurement and OGTT for more reliable judgement and how to minimize discordance between these two major diagnostic tests. 

Overall, this is a well-written paper. 

Author Response

Thank you.